# Conjunctive Analyses of BSA-Seq and BSR-Seq to Identify Candidate Genes Controlling the Black Lemma and Pericarp Trait in Barley

**DOI:** 10.3390/ijms24119473

**Published:** 2023-05-30

**Authors:** Yajie Liu, Pengzheng Chen, Wenshuo Li, Xinchun Liu, Guowu Yu, Hui Zhao, Shuhua Zeng, Mao Li, Genlou Sun, Zongyun Feng

**Affiliations:** 1State Key Laboratory of Crop Gene Exploration and Utilization in Southwest China/College of Agronomy, Sichuan Agricultural University, Chengdu 611130, China; liuyajie@sicau.edu.cn (Y.L.); 13320908732@139.com (P.C.); lws970612@163.com (W.L.); ku0082005@126.com (X.L.); 2002ygw@163.com (G.Y.); zhaohui_hbu@126.com (H.Z.); zshgsp@163.com (S.Z.); tongshengmaoyu@sina.com (M.L.); 2Department of Biology, Saint Mary’s University, Halifax, NS B3H 3C3, Canada

**Keywords:** allomelanin, high throughput sequencing, bulk segregation analysis, benzoic acids derivates, barley

## Abstract

Black barley seeds are a health-beneficial diet resource because of their special chemical composition and antioxidant properties. The black lemma and pericarp (BLP) locus was mapped in a genetic interval of 0.807 Mb on chromosome 1H, but its genetic basis remains unknown. In this study, targeted metabolomics and conjunctive analyses of BSA-seq and BSR-seq were used to identify candidate genes of BLP and the precursors of black pigments. The results revealed that five candidate genes, purple acid phosphatase, 3-ketoacyl-CoA synthase 11, coiled-coil domain-containing protein 167, subtilisin-like protease, and caffeic acid-*O*-methyltransferase, of the BLP locus were identified in the 10.12 Mb location region on the 1H chromosome after differential expression analysis, and 17 differential metabolites, including the precursor and repeating unit of allomelanin, were accumulated in the late mike stage of black barley. Phenol nitrogen-free precursors such as catechol (protocatechuic aldehyde) or catecholic acids (caffeic, protocatechuic, and gallic acids) may promote black pigmentation. *BLP* can manipulate the accumulation of benzoic acid derivatives (salicylic acid, 2,4-dihydroxybenzoic acid, gallic acid, gentisic acid, protocatechuic acid, syringic acid, vanillic acid, protocatechuic aldehyde, and syringaldehyde) through the shikimate/chorismite pathway other than the phenylalanine pathway and alter the metabolism of the phenylpropanoid-monolignol branch. Collectively, it is reasonable to infer that black pigmentation in barley is due to allomelanin biosynthesis in the lemma and pericarp, and *BLP* regulates melanogenesis by manipulating the biosynthesis of its precursors.

## 1. Introduction

Barley (*Hordeum vulgare* L. ssp. *vulgare*) is the fourth most abundant cereal worldwide. It is an adaptable and important species, particularly in regions where the climate and soil are suboptimal for agriculture. Barley grains are rich in dietary fiber (such as β-glucan), secondary plant compounds (such as flavonoids), and tocols, which reduce the risk of diseases and metabolic disorders [1,2,3,4,5]. Its unique chemical composition and health benefits make barley a potentially healthy and functional food. Colored barley has become a popular health food owing to its important free radical scavenging and antioxidant capacity with phenolic acids, anthocyanins, and melanin accumulation [6,7]. Barley is classified into yellow, purple, red, blue, black, and gray types based on the accumulation of different pigments. Unlike other colored barleys, the black and gray colors are due to the accumulation of melanin in the lemma and pericarp [8,9]. Significant amounts of anthocyanins and lignin have also been found in black seed barley [10].

Melanin is a high-molecular-weight pigment synthesized from phenolic precursors, which are oxidized through the action of polyphenol oxidase (PPO) into quinone, followed by its polymerization [11,12]. Melanin is currently classified into eumelanins, pheomelanins, and allomelanins based on their monomers and biosynthesis modes. Allomelanin is synthesized from various nitrogen-free precursors in plants, such as catechol, caffeic acid, chlorogenic, protocatechuic, and gallic acids [11,12,13], while both eumelanins and pheomelanins are derivatives of tyrosine in animals. Allomelanin is a dark, insoluble pigment that leads to brown and black seed color in plants. Despite its biomedical and technological applications as a unique material, natural melanin also has a wide range of biological activities and human health benefits, such as antioxidant, antimicrobial, anti-HIV, anticancer, anti-inflammatory, immunomodulatory, and anti-aging effects [14,15,16,17]. However, the biochemical and genetic aspects of melanin formation in plants have not been extensively studied because of the complex polymeric nature of melanin and the interference of other pigments. Barley black seeds are rich in flavonoid compounds and have a free radical scavenging capacity, which is considered to be the result of environmental adaptation to biotic and abiotic stresses [6,7]. It has been reported that the black seed trait is controlled by a single dominant locus on chromosome 1HL, which manipulates melanin accumulation within senescing plastids in the pericarp and husk tissues of barley [18,19,20,21,22]. Most recently, using the bulked segregant analysis (BSA) method and genetic mapping, the *BLP* gene was mapped to two overlapping genetic intervals of 1.66 and 0.807 Mb on chromosome 1H [23,24]. The metabolic pathways functioning in melanin-accumulating barley grains were characterized by comparative transcriptomic and metabolomic analyses, and the results showed that genes encoding enzymes of the general phenylpropanoid pathway, together with genes of monolignol synthesis, were transcriptionally activated, but none of the key genes in flavonoid biosynthesis was altered at the transcriptional level in black seed barley. Despite the enhanced total amount of phenolic compounds, the biosynthesis of melanin is accompanied by the redirection of metabolic fluxes in the phenylpropanoid pathway, and one potential precursor was revealed in black seed barley in previous work [12,25]. It needs to be clarified whether *BLP* activates the shikimate pathway, which provides precursors for melanin synthesis and phenylpropanoid pathways.

Bulk segregation analysis (BSA) utilizes a pool of individuals with extreme phenotypes to map quantitative trait loci (QTL) [26,27]. With the rapid development of next-generation sequencing (NGS) technologies, the ever-decreasing cost of sequencing has accelerated the development and improvement of BSA methods for different species [26]. Moreover, bulked segregant RNA-seq (BSR) was developed based on the BSA method [28,29]. BSR utilizes transcriptome data from two pools with extreme traits in the segregating population for single nucleotide polymorphism (SNP) marker development and QTL mapping instead of resequencing in BSA. The gene expression information of parents or mixed pools can be used in BSR, which helps screen candidate genes based on their expression differences. However, BSR analysis generally requires a high-quality reference genome, which reduces its efficiency in species with complex genomes. Thus, the combination of BSA and BSR takes advantage of both methods to improve the accuracy and efficiency of mapping [30,31,32].

The objective of this study was to identify candidate genes for the BLP locus using conjunctive analyses of BSA-seq and BSR-seq. Our results revealed five differentially expressed genes on the 1H chromosome as candidate genes for the BLP locus. Targeted metabolomic analysis showed that 17 differential metabolites, including the precursors and repeating units of melanin, accumulated in the grain-filling seeds of black barley. These results provide useful information for further revealing the genetic mechanisms controlling BLP in barley.

## 2. Results

### 2.1. Phenolic Compounds Content Measurements

To identify the precursors for melanin synthesis, 130 phenolic compounds were analyzed by UPLC-ESI-MS/MS in the late milk-stage seeds of the two progeny pools with extreme traits and their parents. The results revealed that 61 metabolites were detected and 17 differential metabolites (DMs) were significantly accumulated in black seed samples (*p*-value ≤ 0.05), including two phenylpropanoids, seven benzoic acid derivatives, four coumarins, and four others, which were enriched in phenylpropanoid biosynthesis, polycyclic aromatic hydrocarbon degradation, plant hormone signal transduction, and the biosynthesis of siderophore group nonribosomal peptides (Table 1). Among the tested phenylpropanoids, caffeic acid and ferulic acid of black seed barley were significantly higher than those of the control, while 4-hydroxycinnamic acid, *p*-coumaric acid, trans-cinnamic acid, sinapic acid, and caftaric acid were also detected without statistically significant differences. Among the tested benzoic acid derivatives, salicylic acid, 2,4-dihydroxybenzoic acid, gallic acid, gentisic acid, protocatechuic acid, syringic acid, and vanillic acid were higher in black seed barley than in the control, whereas 4-hydroxybenzoic acid and 2,6-dihydroxybenzoic acid were also detected without statistically significant differences. Cryptochlorogenic acid, an ester of caffeic acid, protocatechuic aldehyde, and syringaldehyde, as the dedehyde of protocatechuic and syringic acid, was also significantly increased in black seed barley. Notably, caffeic acid, protocatechuic acid, gallic acid, and protocatechuic aldehyde are the major precursors of allomelanin biosynthesis.

### 2.2. Bulked Segregant Analysis Sequencing

#### 2.2.1. Sequencing Data Assessment and Mapping Analysis

To determine the genetic differences between yellow and black barley, high-throughput sequencing was performed using the DNAs of four libraries from two parents and two progeny pools with extreme traits. In total, 1,539,353,156 clean paired-end reads were generated after data filtering. Then, 68.2% of pair-end clean reads on average were mapped to the reference *Hordeum_vulgare* MorexV3 genome (http://plants.ensembl.org/Hordeum_vulgare/Info/Index, accessed on 20 Augest 2022) (Table 2). The genome coverage rate was between 87% and 95.14%, with an average sequencing depth of 6.44 for parents and 32.71 for progeny pools.

#### 2.2.2. Location of Candidate Regions of BSA-Seq Data

A total of 3,305,875 SNPs and 226,428 InDels with different genotypes in parents were selected for association analysis (coverage depth > 5 for parents, coverage depth > 10 for offspring bulks) using the Genome Analysis ToolKit. Of these, 57.69% of the SNP variants were transitions. Of the InDel variants, 85.35% were small indels less than 3 bp, and over half of the InDel variants were 1 bp in length. The variant numbers on each chromosome were generally but not equally distributed (Figure 1). A total of 67,253 SNPs and 8812 InDels were predicted to be located in the genic region, including stopgain, stoploss, and synonymous and non-synonymous mutations. Association analysis was conducted with SNPs and InDels using QTLseqr (V0.7.0; R package). The candidate region for black seed was limited to 10.12 Mb on chromosome 1H with Δ(SNP-index) above the threshold, including 283 genes (Figure 2, Table 3).

### 2.3. Bulked Segregant RNA-Seq Analysis

#### 2.3.1. Sequencing Data Analysis of two cDNA Bulks

To investigate the transcriptional profile of the black seed phenotype, bulked segregant RNA-seq (BSR-seq) analysis was performed with two progeny bulks (F2Y and F2B) using the Illumina HiSeq platform. In total, 23,962,473,126 clean bases were obtained. The two F2 segregation bulks (F2Y and F2B) generated 35,950,689 and 44,476,127 clean reads, respectively (Table 4). The Q30 of the two progeny sequencing bulks was over 92.56%, indicating high quality. Clean reads were assembled according to the reference genome. A total of 29,590 transcripts were obtained, including 7443 novel transcripts (2618 transcripts from 2096 known genes and 4825 transcripts from novel genes), and 50.81% of the novel transcripts were predicted to be lncRNA. A total of 31,879 SNPs and 411 InDels with different genotypes in the parents were selected for association analysis (coverage depth >30 for offspring bulks) using the Genome Analysis ToolKit. In total, 60.92% of the SNP variants were transitions. Association analysis revealed that the candidate region for black seed was limited to 16.26 Mb on chromosome 1H with Δ(SNP-index) above the threshold from cDNA data, which covers the candidate region of BSA analysis (Table 3).

#### 2.3.2. Gene Ontology and Kyoto Encyclopedia of Genes and Genomes Pathway Enrichment Analysis of Differentially Expressed Genes

DEGs between the two F2 segregation bulk samples were screened with |Log_2_(Foldchange)| ≥ 1 and FDR ≤ 0.05, and 37 DEGs were up-regulated and 58 DEGs were down-regulated (Figure 3). The annotation of all DEGs was carried out using the GO, KEGG, EuKaryotic Ortholog Groups (KOG), and SWISS-PROT public databases. A total of 95 DEGs were assigned to 29 KEGG pathways, of which seven pathways were significantly enriched (*p*-value < 0.05), including glycolysis/gluconeogenesis, arginine and proline metabolism, cutin, suberine and wax biosynthesis, beta-alanine metabolism, glycerolipid metabolism, flavone and flavonol biosynthesis, and limonene and pinene degradation (Figure 4). Swiss-Prot annotation revealed that the products of 44 DEGs have enzyme activities of transferase, oxygenase, kinase, esterase, and others associated with the metabolism and biosynthesis of glucose, lipid, quinone, flavonoid, and lignin.

Key enzymes in melanin biosynthesis and phenylpropanoid-monolignol branch biosynthesis were identified in the DEGs. One DEG, named polyphenol oxidase/tyrosinase (HORVU.MOREX.r3.4HG0418160), involved in the melanin pigment biosynthetic process (GO:0046148) and tyrosine metabolism (KO00350), was up-regulated 7-fold in black seed barley. Three DEGs with o-methyltransferase activity and one DEG with esterase activity participated in the metabolism of lignin, named caffeic acid-*O*-methyltransferase (COMT, HORVU.MOREX.r3.1HG0089900 down-regulated), tricin synthase 1 (HORVU.MOREX.r3.3HG0315930 down-regulated), tricetin 3′,4′,5′-*O*-trimethyltransferase (novel.280 up-regulated), and caffeoyl shikimate esterase (HORVU.MOREX.r3.2HG0134440 up-regulated). These results indicate that the black seed phenotype might be associated with many genes and various metabolic processes, especially the metabolism of lignin, lipids, and melanin.

### 2.4. Combined Analysis of Bulk Segregant Analysis Sequencing and Bulk Segregant RNA Sequencing

The BSA-seq and BSR-seq results were jointly analyzed, which revealed that five candidate genes on chromosome 1H were significantly differentially expressed between the black and yellow barley seed samples, which included three up-regulated and two down-regulated genes (Table 5). The up-regulated gene HORVU.MOREX.r 3.1HG0087150 was annotated as a purple acid phosphatase (PAPs; KOG1378), which belongs to the metallophosphatase (MPP) superfamily. PAPs contain a binuclear metal center and exhibit phosphatase activity, catalyzing the hydrolysis of a wide range of activated phosphoric acid mono- and di-esters and anhydrides. The up-regulated gene HORVU.MOREX.r3.1HG0085470 is described as 3-ketoacyl-CoA synthase 11, and participates in fatty acid elongation and plant–pathogen interaction pathway (KO00062, KO04626) and functions in fatty acid biosynthetic process, integral component of membrane, and transfer of acyl groups other than amino-acyl groups (GO:0006633, GO:0016021, GO:0016747). The up-regulated gene HORVU.MOREX.r3.1HG0088450 was named coiled-coil domain-containing protein 167 with unknown functions. Additionally, the down-regulated gene HORVU.MOREX.r3.1 HG0089900 was annotated as caffeic acid-*O*-methyltransferase 1, and participates in the flavone and flavonol biosynthesis pathway (KO00944) and functions in lignin, aromatic compound, melatonin, and flavonol biosynthetic processes (GO:0009809, GO:0019438, GO:0030187, GO:0051555). The down-regulated gene HORVU.MOREX.r3.1 HG0089480 was predicted to be a subtilisin-like protease with serine-type endopeptidase activity and proteolysis.

### 2.5. Validation of Quantitative Reverse Transcription PCR for Candidate Genes Related to Pigmentation

The expression patterns of the five candidate genes from the combined analysis of BSA and BSR were characterized using RT-qPCR. The results show that 3-ketoacyl-CoA synthase 11 (HORVU. MOREX. r3.1 HG0085470), purple acid phosphatase (HORVU. MOREX. r3.1 HG0087150), and coiled-coil domain-containing protein 167 (HORVU. MOREX. r3.1 HG0088450) in the black seed barley samples were significantly higher than those in the yellow samples (*p* < 0.01). However, the expression of subtilisin-like protease (HORVU. MOREX. r3.1 HG0089480) and caffeic acid-*O*-methyltransferase (HORVU.MOREX. r3.1 HG0089900) in the black seed barley samples was significantly lower than that in the yellow samples (*p* < 0.05) (Figure 5). The expression of candidate genes between samples was consistent with the BSR-seq results.

## 3. Discussion

Black seed barley is considered to be more drought- and cold-tolerant in the field. Various metabolites with biological activities were accumulated in this special type of barley. To guide the precise breeding of colored barley, gene mapping and cloning of *BLP* have recently attracted significant research interest. It was reported that black seed is controlled by the monogenic locus *BLP* [19,20,21], which is located at two overlapping genetic intervals of 1.66 and 0.807 Mb on chromosome 1HL, and 14 candidate genes display coding sequence variation between black and yellow seed barley [23,24]. A large population, including 433 double haploid lines and 1009 recombinant inbred lines, was used in Long’s mapping; however, it looks like larger populations are still needed for further fine mapping. The joint analysis of BSA-seq and BSR-seq was used in many crops as an efficient and fast mapping method by using the F2 population. In the present study, we narrowed the *BLP* to five candidate genes in a genetic interval of 10.12 Mb on 1H by jointly analyzing BSA-seq and BSR-seq. Even though this interval was much larger than that in the previous report, only five genes were identified after differential expression profiling. The sequencing depth and individual number for each bulk should be increased in the future to improve the efficiency of this conjunctive method in large-genome plants such as barley. Among the candidates, two genes encoding purple acid phosphatase and 3-ketoacyl-CoA synthase 11 are strongly recommended as candidates in this study after consideration of *BLP* location in previous reports [24]. In Long’s reports, 25 SNPs were in the transcriptional regions of purple acid phosphatase, including four missense SNPs, but only 1 SNP was in the 3′ UTR of 3-ketoacyl-CoA synthase 11. Our results showed that 5 SNPs were detected in the 2 kb upstream of ATG and the transcriptional region, including 2 synonymous mutations in exons, 1 SNP in the probable promoter region from 1169 bp to ATG, and 2 SNPs in the 3′ UTR, but there was only 1 SNP in the promoter region of 3-ketoacyl-CoA synthase 11. Thus, the gene coding purple acid phosphatase should be taken as the candidate gene for causing black pigmentation in barley lemma and pericarp by mutations at the transcriptional and protein levels.

In previous reports, phenolic compounds’ levels in the near-isogenic lines of BLP and its control (BW) indicated that total benzoic acid and caffeic acid content were significantly higher in BLP than in BW, where protocatechuic acid, vanillic acid, and total cinnamic acid content had no significant differences [33]. In order to figure out more character compounds in BLP, 130 phenolic compounds in the late milk stage of F2 bulks and their parents were analyzed. 17 differential metabolites (DMs), including 7 benzoic acid derivatives, were significantly accumulated in black seed samples, which is generally consistent with Glagoleva’s result, and more differential compounds of benzoic acid were exposed. However, protocatechuic and vanillic acid were also significantly higher in black seed F2 bulks and its parent than control in our test, which were not differential metabolites in previous work. The maturing stage of the samples and tested methods may explain these inconsistency results. *BLP* may manipulate the accumulation of hydroxybenzoic acid derivatives (salicylic acid, 2,4-dihydroxybenzoic acid, gallic acid, gentisic acid, protocatechuic acid, syringic acid, vanillic acid, protocatechuic aldehyde, and syringaldehyde) through the shikimate/chorismite pathway other than the phenylalanine pathway and alter the metabolism of the phenylpropanoid-monolignol branch (Figure 6). Among the 12 differential phenolic acids and derivatives, caffeic acid, ferulic acid, and cryptochlorogenic acid (4-*O*-caffeoyl quinine) represent hydroxycinnamic acids and their esters, whereas the remaining nine phenolic acids and derivatives, such as protocatechuic acid, protocatechuic aldehyde, salicylic acid, gallic acid, and syringic acid, represent the main hydroxybenzoic acid derivatives [34]. Hydroxybenzoic acid derivatives are derived through the shikimate/chorismite or phenylalanine pathway [35]. Recently, a report revealed that salicylic acid, a hydroxybenzoic acid derivative, was not initiated from phenylalanine in *Arabidopsis thaliana* [36]. As the compound levels of *t*-cinnamic acid, *p*-coumaric acid (4-hydroxycinnamic acid), and 4-hydroxybenzoic acid in black seed barley were unchanged, but phenylalanine initiates phenylpropanoid metabolism by converting it to *t*-cinnamic acid, we speculated that the nine up-regulated hydroxybenzoic acids might be produced from the shikimate/chorismite pathway rather than the phenylalanine pathway. However, the increased levels of caffeic acid and ferulic acid indicated that the biosynthesis of the phenylpropanoid-monolignol branch was also modified, although it did not start with the alteration of *t*-cinnamic acid and *p*-coumaric acid. The results of transcriptomic analysis confirmed that *BLP* significantly affected the expression of genes encoding for enzymes participating in lignin biosynthesis, such as caffeic acid-*O*-methyltransferase, tricin synthase 1, tricetin 3′,4′,5′-*O*-trimethyltransferase, and caffeoyl shikimate esterase. Tricin synthase 1 and tricetin 3′,4′,5′-*O*-trimethyltransferase catalyze the stepwise methylation of tricetin to its 3′-mono- and 3′,5′-dimethyl ethers or 3′,4′,5′-trimethylated ether derivative to produce a new lignin subunit, called tricin, in monocot lignification, while COMT is involved in the synthesis of both S lignin units and tricin [37,38]. Caffeoyl shikimate esterase (CSE) is an enzyme central to the lignin biosynthetic pathway by hydrolyzing caffeoyl shikimate into caffeic acid and shikimate [39], which is also identified as lysophospholipase2 (LYSOPL2) and capable of removing an *O*-acyl chain (*sn*-1/*sn*-2) from phospholipids to produce lysophospholipids 2 (LPLs2) and free fatty acids. In addition, *AtLYSOPL2* can be induced by various biotic and abiotic stressors, including salicylic acid, pathogens, cold, drought, etc. [40,41]. The up-regulated expression of *CSE* and down-regulated expression of *COMT* may also explain the increase in caffeic acid. As *CSE/AtLYSOPL2* can be induced by salicylic acid and other biotic or abiotic stresses [42], it is reasonable to speculate that up-regulation of caffeic acid (allomelanin precursor) could be due to the increased biosynthesis of salicylic acid and *BLP*-regulated pigment synthesis through the accumulation of hydroxybenzoic acid derivatives.

Allomelanins are heterogeneous structural pigments derived from phenol nitrogen-free precursors, such as catechol or catecholic acids (caffeic, chlorogenic, protocatechuic, and gallic acids) or other types of dihydroxybenzenes [11,12,13]. Our results of targeted metabolomics revealed that *BLP* regulates melanogenesis by accumulating catechol precursors and also confirmed that the black pigmentation in barley is due to allomelanin accumulation in the lemma and pericarp. Black pigment accumulation is accompanied by an increase in 12 phenolic acids and derivatives both in F2B bulk and its parent EH, some of which are considered precursors of allomelanins. Six anthocyanin-tested compounds were not detected in black seed barley, while melanin in plants was reported as a nitrogen-free type [43], these suggest that the pigments of black seed barley could be allomelanin and derived from nitrogen-free monomers but not anthocyanins. The results of the transcriptomic analysis confirmed that none of the main enzymes involved in anthocyanin biosynthesis had changed expression at the transcription level. Notably, allomelanin could be derived from the catechol monomer, mainly from caffeic acid and protocatechuic aldehyde, because of the higher content or fold change in all differential metabolites.

Abundant hydroxybenzoic acid derivatives make black seed barley a health-benefit diet and confer good resistance to abiotic and biotic stresses by possessing biological activity and antioxidant properties. Black seeds are rich in flavonoid compounds and have free radical scavenging capacity in barley [6,7], but few typical compounds have been characterized in comparison with yellow seed barley. In this study, 12 phenicol acid derivatives and 4 coumarins with high medicinal value were detected in black seed barley, such as caffeic acid, which has strong antioxidant, antifungal, anti-inflammatory, and anticancer properties [44]. Black-seeded barley types demonstrated higher resistance to drought, cold, and fungal infections, which might be due to the increased levels of ferulic acid and other phenolic compounds [25,45,46,47]. In this study, increased salicylic acid was suggested as another reason for the higher antioxidant capacity and biotic and abiotic stress tolerance in black seed barley. Salicylic acid (SA) is an important plant hormone, and its accumulation is known to boost the host defense system upon infection with pathogens [48,49,50,51]. The up-regulated expression of *PR1* (*pathogenesis-related protein 1*, HORVU.MOREX.r3.5HG0519330 2.9-fold up-regulated) downstream of the salicylic acid hormone signaling pathway reflects the activation of disease resistance in black seed barley.

Collectively, these results revealed that purple acid phosphatase, 3-ketoacyl-CoA synthase 11, coiled-coil domain-containing protein 167, subtilisin-like protease, and caffeic acid-*O*-methyltransferase in the 10.12 Mb location region on the 1H chromosome were considered candidate genes of BLP by conjunctive analyses of BSA-seq and BSR-seq. 12 phenicol acid derivatives and 4 coumarins with high medicinal value, including the precursor and repeating unit of allomelanin, were detected in black seed barley, which demonstrated a role for BLP in the benzoic acid derivates and lignin biosynthetic pathway. The accumulation of salicylic acid and allomelanin precursors suggested that the key point of BLP-manipulated melanogenicity and high resistance is the phenolic acids in the shikimate pathway. However, there is still a lack of evidence to suggest that the *BLP* gene directly manipulates melanin metabolism. Transgenic materials will need to be created to address this issue.

## 4. Materials and Methods

### 4.1. Plant Materials

The Chinese hulless barley variety Zhongnuo8 (ZB, yellow lemma and pericarp) and barley variety Erlenghei (EH, black lemma and pericarp) were crossed to generate F1 (Figure 7). An F2 population of 1022 lines derived from F1 was planted in the Yangma experimental field in Chengdu, China. The grain color (according to the lemma and pericarp) of the F2 populations was checked in soft-hard dough stage seeds (Figure 7). Thirty individuals with extreme phenotypes were selected according to phenotype assessment. The whole seeds (with glumellae) in the late milk stage from the middle part of the spike of the selected plants were mixed (one seed per individual) and stored in liquid nitrogen immediately after harvest for further DNA and RNA extraction and chromatographic analysis.

### 4.2. Extraction and Chromatographic Analysis of Phenolic Compounds

Samples of parents (ZB and EH) and F2 segregation bulk samples (F2Y and F2B, 30 individuals per bulk) were analyzed using ultra-performance liquid chromatography-electrospray ionization-tandem mass spectrometry (UPLC-ESI-MS/MS). Whole grains were immediately mixed and frozen in liquid nitrogen after harvest. The profiles of 130 phenolic compounds, which were classified into anthocyanins, benzoic acid derivatives, catechin derivatives, coumarin, dihydrochalcones, flavanones, flavones, flavonols, isoflavones, phenylpropanoids, proanthocyanidins, stilbenes, and terpenoids, were analyzed by Shanghai Lu-Ming Biotech Co., Ltd. (Shanghai, China). Three independent extraction procedures were conducted for each sample. MeOH:water (2:1, *v*/*v*, containing IS) and ultrasonic were repeated for metabolite extraction, and then all the supernatant was dried and re-dissolved in 200 μL of MeOH:water (7:18, *v*/*v*, containing IS) and filtered through a 0.22 μm organic phase pinhole filter for UPLC-MS/MS analysis. Separation was implemented on a Waters UPLC HSS T3 (100 × 2.1 mm, 1.8 μm) column in a binary solvent system consisting of 0.1% methanol in water and acetonitrile. The injection volume was 5 µL, the column temperature was 40 °C, and the flow rate was 0.35 mL/min. Mass spectrometry was performed on the API 6500+ Qtrap System (AB SCIEX, Framingham, MA, USA) with an electrospray ionization source using the following parameters: positive ion mode: CUR: 35 Psi; EP: 10 V; IS: 5500 V; CXP: 10 V; TEM: 500 °C; Gas1: 60 Psi; Gas2: 50 Psi; negative ion mode: CUR: 35 Psi; EP: −10 V; IS: −4500 V; CXP: −20 V; TEM: 500 °C; Gas1: 60 Psi; Gas2: 50 Psi.

### 4.3. Bulked Segregant Analysis Sequencing

Genomic DNA was extracted from the seeds of parental materials (ZB and EH) and extreme individuals in the F2 population using the cetyltrimethylammonium bromide (CTAB) method [52]. The DNA concentration was measured using a nanodrop spectrophotometer and agarose gel electrophoresis. DNA sequencing of the bulks and parents was conducted by Wuhan Genoseq Technology Co., Ltd. (Wuhan, China) using an Illumina HiSeq sequencing platform. After filtering the raw reads, BWA software (version 0.7.15-r1140) was used to map the clean reads to the reference genome (MorexV3; http://plants.ensembl.org/Hordeum_vulgare; accessed on 20 Augest 2022), and SAMtools (version 1.3.1) was used for trans-format, data-sort, and duplicate-removal. SNPs and InDels with different genotypes were screened and annotated using the Genome Analysis ToolKit (GATK, version 3.7) and ANNOVAR (version 2016 Feb1). The association analysis was conducted with SNPs and InDels using the QTLseqr (V0.7.0; R package) with the Δ(SNP-index) method.

### 4.4. Bulked Segregant RNA Sequencing

The total RNA of whole seeds (with glumellae) in the late milk stage from the two bulks with extreme phenotypes from the F2 population was extracted using the Trizol method (DP424, TIANGEN Co., Beijing, China) with a minor modification, adding a step as centrifugation of the mixture of seed powder and trizol and only using the supernatant for RNA extraction. RNA quality was confirmed using a LabChip GX Touch 24 nucleic acid analyzer (RNA quality score ≥ 7.1 for all samples; Appendix A). After quality control, sequencing was performed using an Illumina HiSeq sequencing platform by Wuhan Genoseq Technology Co., Ltd. (Wuhan, China). Fastp software (version 0.23.0) was used to remove adapters and low-quality sequences from the raw data to obtain data with read lengths over 50 bp. Clean data were mapped to the reference genome (MorexV3) using Hisat2 software (version 2.1.0) with the BWT and Ferragina–Manzini (Fm) index methods. Data sorting and file trans-format were conducted using the SAMtools software (version 1.3.1). The screening of SNPs and InDels with different genotypes and association analysis were conducted using the same method described for bulked segregant analysis sequencing.

### 4.5. Analysis of Differential Expression Genes and its Association Analysis with BSA

After normalization, differential expression analysis was performed using DESeq2 v1.10.1. Differentially expressed genes (DEGs) were identified between F2Y and F2B samples with |Log_2_(Foldchange)| ≥ 1 and FDR ≤ 0.05. The R package (hypergeometric test) was used to perform annotation and enrichment analysis of the Gene Ontology (GO) and Kyoto Encyclopedia of Genes and Genomes (KEGG) of DEGs. Significantly enriched pathways were screened with a Q-value ≤ 0.05. The candidate loci identified by BSA-seq and BSR-seq were compared with DEGs. Genes with significantly different expression in the location regions were considered candidates for *BLP*.

### 4.6. Real-Time Quantitative PCR Validation

The expression of candidate genes was validated by quantitative reverse transcription PCR (RT-qPCR). Powdered whole seeds (100 mg, with glumellae) in the late milk stage from the F2 segregation bulk samples (F2Y and F2B) were used for the extraction of total RNA using the Trizol method with three biological replicates. Total RNA (1 μg) was used for cDNA synthesis using FastKing gDNA Dispelling RT SuperMix (TIANGEN Co., China) following the manufacturer’s instructions. All primers were designed using Primer 5 and are listed in Appendix A. RT-qPCR assays were performed using SuperReal PreMix Color (SYBR Green) (TIANGEN Co., China) by a LightCycler^®^ 96 Instrument (ROCHE Ltd., Shanghai, China) with three technical replicates in a 20 μL reaction mixture according to the manufacturer’s instructions. The relative expression levels of the target genes were calculated using the 2^−ΔΔCt^ method. GAPDH was used as the internal reference gene.

## Figures and Tables

**Figure 1 ijms-24-09473-f001:**
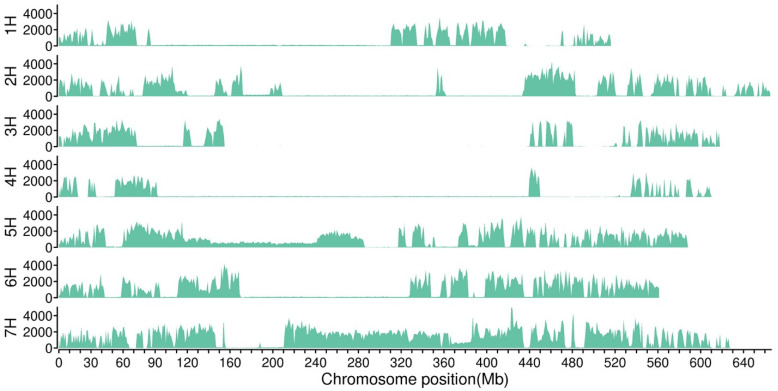
Variant distribution on each chromosome.

**Figure 2 ijms-24-09473-f002:**
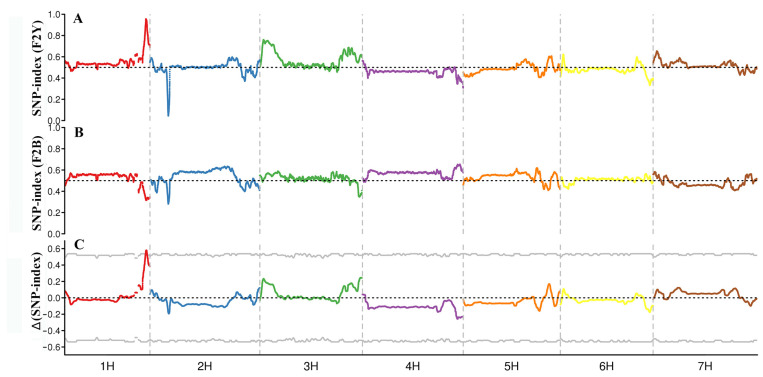
Distribution of SNP index association values on chromosomes. (**A**,**B**). Distribution of single nucleotide polymorphism (SNP) index values of F2Y (**A**) and F2B (**B**). (**C**) Distribution of Δ (SNP index) value on chromosomes, where the grey line represents the threshold (*p* = 0.001).

**Figure 3 ijms-24-09473-f003:**
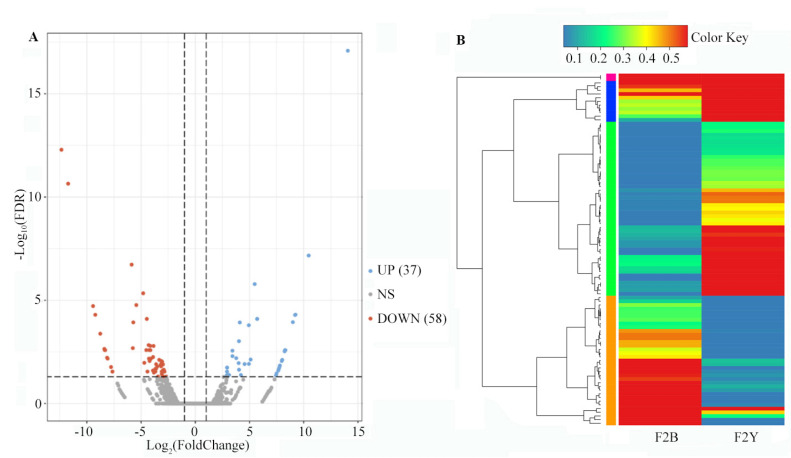
Volcano plots and hierarchical clustering map of differentially expressed genes (DEGs) between F2B and F2Y bulk in BSR-seq analysis. (**A**). Volcano plots based on log_2_FC (F2B/F2Y) and FDR values. Red scatter points represented the upregulated genes, blue scatter points represented the down-regulated genes. (**B**). Hierarchical clustering map based on TPM value.

**Figure 4 ijms-24-09473-f004:**
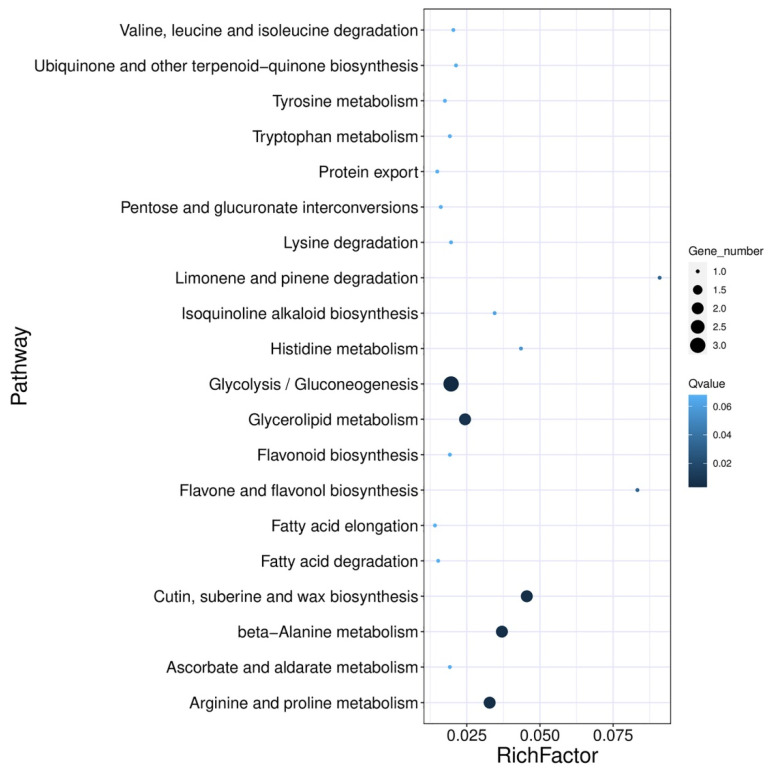
KEGG pathway enrichment of differential expression gene among F2B and F2Y bulk.

**Figure 5 ijms-24-09473-f005:**
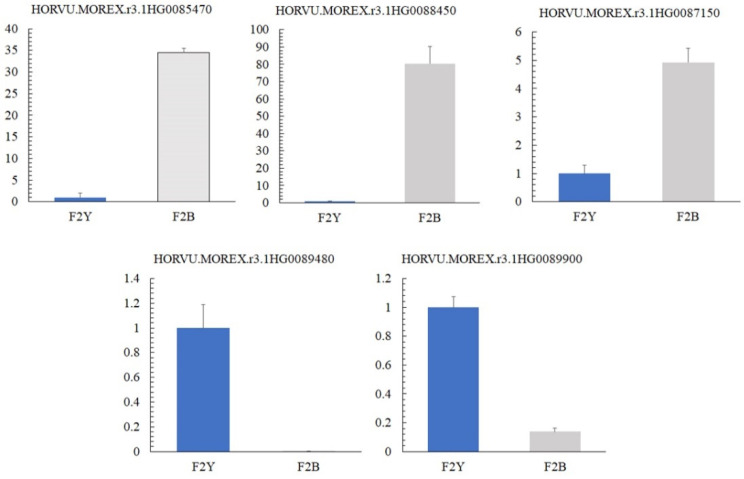
Expression validation of candidate genes by RT-qPCR. The actin gene was used as an internal control. Error bars represent the mean SE of three biological replicates.

**Figure 6 ijms-24-09473-f006:**
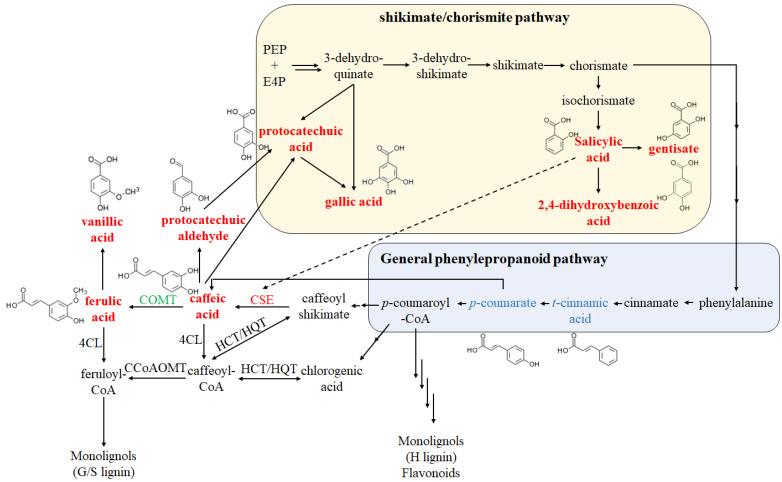
The biosynthetic network for benzoic acid derivatives. Red and green characters indicated the up- and down-regulated compounds/genes in black-seed barley, respectively, blue characters indicated the detected compounds without significant differences in the two comparison samples. Abbreviation: PEP, phosphoenolpyruvate; E4P, D-erythrose 4-phosphate; COMT, caffeic acid *O*-methyltransferase; CSE, caffeoyl shikimate esterase; HCT/HQT, hydroxycinnamoyl-CoA shikimate/quinate hydroxycinnamoyl transferase.

**Figure 7 ijms-24-09473-f007:**
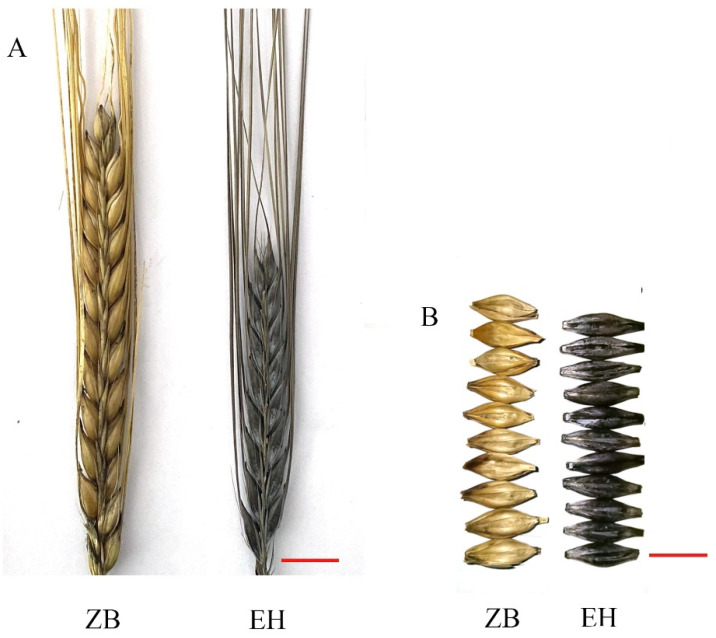
Phenotype of mature seeds in parent lines. Spike (**A**) and seed (**B**) of the ZB and EH. Bars = 1 cm.

**Table 1 ijms-24-09473-t001:** The content of differential metabolites in comparison of parents (ZB, EH) and F2 segregation bulks (F2Y, F2B).

Metabolites	ZB	EH	F2Y (ng/g)	F2B (ng/g)	Category
	(ng/g)	(ng/g)			
Caffeic acid	205.67	545.79	362.01	1956.38	Phenylpropanoids
Ferulic acid	1056.49	2046.86	2877.01	4901.19	Phenylpropanoids
2,4-dihydroxybenzoic acid	4.75	23.20	42.79	84.32	Benzoic acid derivatives
Gallic acid	3.08	5.12	2.30	11.06	Benzoic acid derivatives
Gentisic acid	5.24	21.02	45.00	82.99	Benzoic acid derivatives
Protocatechuic acid	39.37	264.94	75.93	422.88	Benzoic acid derivatives
Salicylic acid	11.54	121.50	27.16	79.77	Benzoic acid derivatives
Syringic acid	147.69	285.26	299.53	517.16	Benzoic acid derivatives
Vanillic acid	1101.53	1538.71	1750.13	3492.70	Benzoic acid derivatives
Cryptochlorogenic acid	26.89	147.85	64.98	1161.05	Alcohols and polyols
Protocatechuic aldehyde	13.78	173.09	12.41	207.28	Catechols
Syringaldehyde	14.15	49.05	17.05	65.25	Benzaldehydes
Narcissin	69.76	196.26	227.58	333.83	Flavonols
3,4-dihydrocoumarin	0.00	1.11	0.00	1.04	Coumarins
Aesculetin	1.93	12.06	9.41	84.72	Coumarins
Aesculin	4.77	10.57	30.62	58.08	Coumarins
Daphnetin	1.43	14.93	8.66	91.28	Coumarins

**Table 2 ijms-24-09473-t002:** Summary of sequencing data and the alignment result of BSA-seq.

Sample	Total Reads	Mapping Rate (%)	Q30 (%)	Coverage (%)	Average Depth (×)
ZN	169,059,857	56.86275128	94.85	87	6.54
EH	188,406,475	48.39110174	93.94	87.94	6.3
F2Y	705,192,609	88.6376719	95.56	95.14	40.53
F2B	476,694,215	78.9104508	93.09	95.1	24.89

**Table 3 ijms-24-09473-t003:** Candidate genomic regions identified by BSA-seq.

Method	Chromosome	Start (Mb)	End (Mb)	Length (Mb)	Gene Numbers
BSA SNP-index	Chromosome 1H	494.45	504.58	10.12	283
BSR SNP-index	Chromosome 1H	490.29	506.55	16.26	455

**Table 4 ijms-24-09473-t004:** Summary of sequencing data and the alignment result of BSR-seq.

Sample	Total Reads	Mapping Rate (%)	Q30 (%)	Coverage (%)	Average Depth (×)
F2Y	35,950,689	88.19726932	92.56	2.59	83.99
F2B	44,476,127	87.70702314	93.71	2.72	98.27

**Table 5 ijms-24-09473-t005:** The information of candidate genes from combined analysis of BSA and BSR.

Gene_ID	Start	End	Functional Annotations	Num. of SNP
HORVU.MOREX.r3.1HG0085470	495,853,418	495,855,011	3-ketoacyl-CoA synthase 11	1
HORVU.MOREX.r3.1HG0087150	499,006,879	499,010,239	Purple acid phosphatase	5
HORVU.MOREX.r3.1HG0088450	500,843,053	500,843,565	Coiled-coil domain-containing protein 167	1
HORVU.MOREX.r3.1HG0089480	503,466,785	503,471,314	Subtilisin-like protease	2
HORVU.MOREX.r3.1HG0089900	504,155,100	504,157,234	Caffeic acid-*O*-methyltransferase	6

## Data Availability

The data presented in this study are available upon request from the corresponding author.

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
