# Peer review of "Conjunctive Analyses of BSA-Seq and BSR-Seq to Identify Candidate Genes Controlling the Black Lemma and Pericarp Trait in Barley"

_ijms, 2023, doi:10.3390/ijms24119473_

Round 1

Reviewer 1 Report

Regarding the methods section: RNA-Extraction from milk ripe kernels is not trivial. Therefore it is desirable to give quality scores for the extracted RNA if possible. Furthermore, it is desirable to add information about the treatment of the samples (regarding the conservation of RNA) from harvest to extraction. This was already done in the section "1.1. Extraction and Chromatographic Analysis of phenolic compounds", but whether the samples were frozen directly after the harvest remains unclear.

The writing in the discussion section needs to be reworked, mainly regarding orthography. Also, in the sentence "Key enzymes in melanin bio-synthesis and phenylpropanoid-monolignol branch biosynthesis were identified in the.", the final word "DEGs" is missing/misplaced.

Reviewer 2 Report

Reviewer’s Comments

This manuscript investigates the Conjunctive Analyses of BSA-seq and BSR-seq to Identify Candidate Genes Controlling the Black Lemma and Pericarp Trait in Barley. Although the topic of manuscript is interesting and under the scope of this journal, however, extensive improvement is needed before acceptance. So, based on my evaluation, I recommend revisions to authors.

Please always provide the line numbering throughout the manuscript to ease-up the review!!!

Why it is worth full to find out the reason of black Lemma and Pericarp in barley? Is it correlated with production, taste, odor or something that is significant?

Abstract: Name these five candidate genes. And how can you propose that those were the only genes for this color?

Please further explain the concentrations and volumes of the compounds used for qPCR to duplicate this study.

Why GAPDH was used as a reference gene instead of Actin?

It is not common to use references in the results sections.

The conclusion section should be improved.

Reviewer 3 Report

The authors decided to figure out the molecular background of the color variation in the black barley. They performed BSA-seq and BSR-seq coupled with metabolomic analysis and they found several candidate genes the altered expression of which could be the reason for the observed phenotype. The paper is well-written, although there are some parts that should be fixed (listed below). The figures are nice and understandable. I have only a side note: I think the authors should have searched for transcription factors as well, not just enzymes. Maybe a less strict filtering rule (i.e. filter only for the significance without filtering for fold change) would have resulted in a bigger list containing more interesting transcription factors.

Altogether, I recommend accepting this paper after a few modifications.

Rewrite the following part in the Abstract because it is messy: "The results revealed that five candidate genes, purple acid phosphatase, 3-ketoacyl-CoA synthase 11, coiled-coil domain-containing protein 167, subtilisin-like protease and caffeic acid-O-methyltransferase, of the BLP locus were identified in the 10.12 Mb location region on the 1H chromosome after differential expression analysis, and 17 differential metabolites, including the precursor and repeating unit of allomelanin, were accumulated in the late mike stage of black barley. Phenol nitrogen-free "

In page 7, change the word "jointly" to "joint", delete the sentence "It’s worthy to try it in barley for gene mapping", and change the word "maybe" to "may", and the word "inconsistent" to "inconsistency"
